# GWAS Reveals Stable Genetic Loci and Candidate Genes for Grain Protein Content in Wheat

**DOI:** 10.3390/cimb47120981

**Published:** 2025-11-25

**Authors:** Yuxuan Zhao, Renjie Wang, Keling Tu, Yi Hong, Feifei Wang, Juan Zhu, Chao Lv, Rugen Xu, Baojian Guo

**Affiliations:** 1Jiangsu Key Laboratory of Crop Genomics and Molecular Breeding, Agricultural College, Yangzhou University, Yangzhou 225009, China; mx120230803@stu.yzu.edu.cn (Y.Z.); mz120241421@stu.yzu.edu.cn (R.W.); 008443@yzu.edu.cn (K.T.); mz120211269@stu.yzu.edu.cn (Y.H.); feifei.wang@yzu.edu.cn (F.W.); 007670@yzu.edu.cn (J.Z.); clv@yzu.edu.cn (C.L.); 2Key Laboratory of Plant Functional Genomics of the Ministry of Education, Agricultural College, Yangzhou University, Yangzhou 225009, China; 3Jiangsu Key Laboratory of Crop Genetics and Physiology, Agricultural College, Yangzhou University, Yangzhou 225009, China; 4Jiangsu Co-Innovation Center for Modern Production Technology of Grain Crops, Agricultural College, Yangzhou University, Yangzhou 225009, China

**Keywords:** wheat, grain protein content, genome-wide association analysis, grain quality

## Abstract

Grain protein content (GPC) is a key quality trait in wheat, determining both nutritional value and end-use functionality, yet its genetic architecture is complex and highly influenced by the environment. In this study, a diverse panel of 327 wheat accessions was evaluated for GPC across multiple environments. Significant phenotypic variation was observed, with best linear unbiased estimates (BLUEs) ranging from 12.80% to 18.79%, and a moderate broad-sense heritability (*h*^2^ = 0.52) was estimated. Genotype-by-environment interactions were highly significant. Genome-wide association analysis using the FarmCPU model identified seven stable quantitative trait nucleotides (QTNs) associated with GPC on chromosomes 1A, 1B, 2A, 2D, 3B, 5A, and 6A. Among these, *QGpc.yzu-2A* was consistently detected in three environments. Further analysis of the *QGpc.yzu-2A* region identified 26 annotated genes, 8 of which were expressed in grains. One gene, *TraesCS2A02G473000* (RNA-binding protein), exhibited high nucleotide diversity and is a strong candidate for functional validation. Additionally, *QGpc.yzu-6A* co-localized with the known *TaNAM-6A* gene, reinforcing the role of this region in GPC regulation. This study provides valuable insights into the genetic basis of GPC in wheat and offers molecular markers and candidate genes for marker-assisted selection to improve grain protein content in breeding programs.

## 1. Introduction

Bread wheat (*Triticum aestivum* L.), a staple crop for over a third of the world’s population, is a primary source of dietary protein and calories [1,2]. Grain protein content (GPC) is a paramount quality trait that directly defines the nutritional value for human consumption and livestock feed. Moreover, GPC is a critical determinant of functional properties for end-use processing, influencing dough strength, elasticity, and the baking quality of flour [3]. Therefore, genetic enhancement of GPC is a major breeding objective aimed at improving the overall quality and economic value of wheat.

The genetic improvement of GPC is notoriously challenging. It is a complex quantitative trait controlled by multiple genes with small effects, heavily influenced by environmental factors such as nitrogen availability and water stress, and it frequently exhibits a strong negative correlation with grain yield [4]. Conventional phenotypic selection for GPC is inefficient due to its high cost, low heritability, and significant genotype-by-environment (G × E) interactions. This underscores the critical importance of elucidating its genetic architecture. Precise identification of genomic regions and candidate genes governing GPC is fundamental for breaking the yield–protein trade-off and accelerating breeding cycles through marker-assisted selection (MAS) and genomic selection (GS) [5,6].

Genome-wide association studies (GWASs) have been successfully applied to dissect the genetic basis of GPC in wheat, identifying numerous significant marker–trait associations (MTAs) and revealing the complex and polygenic nature of this trait [3,7]. For instance, employing a diverse population of 255 accessions from 27 countries across 5 continents, Kartseva et al. revealed novel, stable genomic regions on chromosomes 3A and 3B that harbor markers associated with GPC [8]. Another study using 93 spring common wheat varieties and breeding lines adapted to Siberia, Russia, reported 11 genomic regions associated with GPC. Nine of these were physically mapped to a region on chromosome 6A that harbors the NAM-A1 gene, a homeolog of the well-known Gpc-B1 (NAM-B1) gene [9,10,11]. Recently, Kartseva et al. performed GWAS for GPC and identified 16 significant QTNs in wheat across the three environments and the BLUEs dataset. These QTNs were distributed on 11 chromosomes [12].

For this study, we evaluated GPC in a diverse panel of 327 wheat accessions. Our analyses revealed significant genetic variation in grain quality traits. Association mapping using high-density SNP arrays indicated a quantitative genetic architecture, with several loci significantly associated with GPC. By leveraging the wheat reference genome and genomic diversity data from wheat pangenome initiatives, we identified putative candidate genes underlying GPC inheritance. These findings provide valuable genetic resources for breeders to enhance GPC and nutritional quality of wheat and facilitate the development of closely linked markers for molecular breeding of improved varieties.

## 2. Materials and Methods

### 2.1. Plant Material

The plant material consisted of 327 wheat accessions, most of them originating from China (297/327, 90.83%), of which 162 were modern varieties and 165 were landrace (historic varieties of tall stature and landraces) [13]. All wheat accessions were planted at the Zhenjiang and Yancheng site with cropping cycles from November to May. The trials were performed in the 2023–2024 and 2024–2025 cropping seasons in Jiangsu Province. We illustrate daily mean temperature and monthly rainfall patterns in Zhenjiang and Yancheng across two growing seasons (November–May of 2023–2024 and 2024–2025) (Appendix A). Both sites exhibited characteristic seasonal temperature progression, with Yancheng showing slightly cooler winter conditions. Rainfall distribution differed regionally, with Zhenjiang displaying more pronounced monthly precipitation peaks compared to Yancheng, reflecting local climatic variations relevant to agricultural planning. Each genotype was sewn in 1 m long row, each with a 20 cm row-to-row spacing. The total nitrogen (N) application rate was 225 kg·ha^−2^, with 50%, 30%, and 20% applied as base fertilizer, tillering fertilizer, and jointing fertilizer, respectively. The application rates of phosphorus (P_2_O_5_) and potassium (K_2_O) were both 144 kg·ha^−2^, with 50% applied as base fertilizer and 50% as jointing fertilizer.

### 2.2. Phenotyping

A total of 327 accessions were evaluated for grain protein content (GPC). The phenotypic data of the GPC were collected from four environments, with each environment representing a combination of location and year. Two to three replications were set up per environment. The wheat GPC was evaluated using a standard near-infrared measurement. The phenotypic measurements were carried out by using sample volumes of 30 g grains per harvested field plot and an Infratec™ 1241 Grain Analyzer (Foss Tecator Co., Ltd., Hillerod, Denmark) applying wavelengths of 730–1100 nm.

### 2.3. Statistical Analyses

An analysis of variance (ANOVA) was applied to test the significance of differences in GPC attributable to accessions (genotype), growing seasons (environment), and their interaction (G × E). The rationale for using an ANOVA was its capability to deconstruct the total observed phenotypic variance into these key components, providing an F-test to determine if the variations introduced by each factor are significantly greater than the residual variance. Subsequently, the best linear unbiased estimators (BLUEs) for each accession across multiple growing seasons were computed. The objective was to derive a robust, adjusted mean value for each genotype that is minimally influenced by the specific conditions of any single season. To achieve this, a mixed linear model was employed, wherein genotype was set as a fixed effect to obtain precise estimates for the defined set of accessions in our panel, and growing season was set as a random effect. This model structure effectively accounts for the random sampling of environmental conditions (seasons) and allows for the generalization of findings beyond the specific years of testing, thus providing reliable genotypic values for downstream correlation and heritability analyses. Pearson’s correlation coefficients (r) were then calculated to evaluate the consistency of GPC performance across different growing seasons and the association between individual season data and the BLUEs.

Broad-sense heritability (*h*^2^) was estimated to assess the degree to which GPC is under genetic control, where *V_G_* is the genotypic variance and *V_E_* is the environmental variance. This estimation is based on variance components derived from the ANOVA and reflects the proportion of total phenotypic variance that is heritable, indicating the potential for phenotypic selection on this trait. Broad-sense heritability *h*^2^ for GPC was calculated using the following formula:$$h^{2}=\frac{V_{G}}{V_{G}+V_{E}}\times 100\%$$

All statistical computations were conducted using the Statistical Package for the Social Sciences (SPSS) version 29.0 (IBM Corp., Armonk, NY, USA).

### 2.4. Association Mapping and Candidate Gene Search

Before performing marker–trait association analysis, the population stratification, the genetic relatedness among population entries, and the LD were considered as in a previous study [13]. The genotypic data for the association panel of 327 accessions were described in the study by Basheir et al. [13]. The values of LD decay (in Mbp) were determined for each chromosome as described in [13]. Hence, the filtered set of 69,441 SNPs, phenotypic data from the four crop seasons, and the calculated BLUE mean values for the traits were obtained using Genomic Association and Prediction Integrated Tool (GAPIT) version 3.0 in R package with a fixed and random Circulating Probability Unification (FarmCPU) model to control pseudo-associations [14]. The FarmCPU model was selected as the sole method for this genome-wide association study due to its demonstrated superiority in effectively controlling false positives while maintaining high statistical power. Unlike the Mixed Linear Model (MLM), which can over-correct for population structure and lead to false negatives, or the fixed-effect model (e.g., stepwise regression), which is prone to false positives due to confounding between population structure and kinship, FarmCPU iteratively employs a fixed-effect model and a random-effect model. This innovative approach effectively separates the testing for marker–trait associations from the control of population structure and kinship [15]. Dashed and solid horizontal lines denote the suggestive threshold value (*p* < 1.0 × 10^−4^) and significance threshold value (*p* < 7.20 × 10^−7^), respectively.

### 2.5. Candidate Gene and Haplotype Analyses in 17 Chinese Wheat Cultivars Leveraging Chromosome-Level Genomes

We identified high-confidence genes and their functional annotations within a 2 Mb region (1 Mb upstream and downstream) surrounding the most significant markers for grain protein content (GPC) on chromosome 2A. To refine the QTL regions and pinpoint putative candidate genes, we examined QTL haplotype structure and nucleotide diversity using genomic resources from the 17 wheat pan-genome [16] and the Chinese Spring reference sequence v1.1. A total of 18 varieties were included in the analysis. Gene sequences derived from Chinese Spring were aligned against the genomes of the 17 varieties via BLAST (http://202.194.139.32/blast/blast.html, accessed 10 September 2025). Subsequent multiple sequence alignment and coding region SNP calling were performed using SnpGene 6.0.2 software. Haplotype number and nucleotide diversity were assessed with DnaSP v5, while gene expression data were obtained from public sources [17].

## 3. Results

### 3.1. Phenotypic Variation

The assessment of 3 wheat grains’ GPC was performed in four environments on a set of 327 wheat varieties (Table 1). The range of coefficients of variation (CV, %) across the individual years was similar for the GPC. Estimates of broad-sense heritability (*h*^2^) for GPC ranged from 0.20 to 0.49. To exclude the impact of the growing season, we computed best linear unbiased estimator (BLUE) values for each accession, treating genotype as fixed and growing season as a random effect. BLUEs varied across the years from 12.80 to 18.79%, being on average 15.37% for GPC (Table 1 and Appendix A; Figure 1A–E), and they showed a wide genotypic variation (*h*^2^ = 0.52). To assess the trait consistency across the environments and BLUE values for GPC, Pearson’s correlation coefficient was used. Low to high positive Pearson’s correlation coefficients (r) over the years were computed for GPC (spanning from 0.36 to 0.82).

The Shapiro–Wilk test confirmed that the phenotypic data followed a normal distribution (*p* > 0.05). Subsequently, the ANOVA revealed significant phenotypic variation for GPC, with genotype, environment (growing season), and their interaction all exhibiting highly significant effects (Table 2).

### 3.2. GWAS for Significant Quantitative Trait Nucleotides (QTNs)

With the FarmCPU model at *p* = 1 × 10^−4^, seven significant quantitative trait nucleotides (QTNs) were identified on chromosomes 1A, 1B, 2A, 2D, 3B, 5A and 6A (Figure 2A,B), and variation in the QTNs was associated with higher and lower GPC at mature grains (Figure 2C–I). To identify candidate genes regulating grain protein content via GWAS, a ~2-Mb (1-Mb upstream and downstream) wheat reference genomic sequence around the most significant markers was retrieved to characterize the QTL physical region. We named them *QGpc.yzu-1A*, *QGpc.yzu-1B*, *QGpc.yzu-2A*, *QGpc.yzu-2D*, *QGpc.yzu-3B*, *QGpc.yzu-5A*, and *QGpc.yzu-6A*. Of them, *QGpc.yzu-2A* was detected in three environments (E3, E4 and BLUE), and others were detected in two environments (Figure 2A,B and Appendix A, Table 3). *QGpc.yzu-1B* had the largest −*log10*(*P*) value and reached 11.7. A total of 146 high-confidence annotated genes were detected around the 7 QTLs according to the wheat reference v2.1 (Appendix A). These genes were evaluated as potential candidate genes. The genes directly hit by the significant SNPs, along with their putative functions, are presented in Appendix A.

### 3.3. Characterization of the Physical Regions of QGpc.yzu-2A Revealed Potential Candidate Genes for Grain Protein Content in Wheat

Further analysis revealed that 26 annotated genes were detected around the *QGpc.yzu-2A* region in the present study. Expression analysis indicated that only eight genes were observed in the grain tissue (Figure 3). Notably, the expression of gene *TraesCS2A02G473000* was markedly abundant in the analyzed tissues, especially in grains, relative to other genes. The haplotype structure and the nucleotide diversity of eight genes were investigated across the 18 sequenced wheat varieties. *TraesCS2A02G471600* and *TraesCS2A02G473200* showed no variation at all. Other genes shared 2~3 haplotypes, and *TraesCS2A02G473100* was absent in three varieties (Abo, XY6 and JM22). Two genes (*TraesCS2A02G473000* and *TraesCS2A02G473100*) showed relatively high nucleotide diversity within the coding region (Table 4 and Appendix A). In particular, both genes were affected by large-effect mutations compared to Chinese Spring, and a splice region variant was identified compared with other varieties.

## 4. Discussion

Grain protein content is a critically important quality trait in wheat, governing both nutritional value and end-use functionality. In this study, we employed a genome-wide association approach to dissect the genetic basis of GPC in a diverse panel of 327 wheat accessions, predominantly of Chinese origin. Our multi-environment trials revealed considerable phenotypic variation for GPC, with BLUE values ranging from 12.80% to 18.79%, consistent with previous reports in wheat [8,11]. The moderate broad-sense heritability (*h*^2^ = 0.52) indicates a substantial genetic component underlying GPC, yet the significant genotype-by-environment (G × E) interaction and variable heritability across seasons (0.20~0.49) highlight the strong modulating effect of environmental factors. These findings align with earlier studies emphasizing the complex quantitative nature of GPC and the challenges it poses for breeding [4,5].

Using the FarmCPU model, we identified seven QTNs associated with GPC. The stability of *QGpc.yzu-2A* across two environments and the major effect of *QGpc.yzu-1B* make them prime targets for breeding applications. In addition, *QGpc.yzu-2A* and *QGpc.yzu-6A* were also detected by the MLM model. To translate these findings into practice, the flanking SNPs of these stable QTNs, particularly *QGpc.yzu-1B*, *QGpc.yzu-2A* and *QGpc.yzu-6A*, can be developed into robust competitive allele-specific PCR markers. These markers would enable efficient marker-assisted selection to pyramid high-GPC alleles into elite breeding lines, thereby breaking the historical yield–protein trade-off. Furthermore, incorporating these QTNs as fixed effects in genomic prediction models could enhance the accuracy of selecting for GPC, especially when breeding for performance stability across diverse environments.

Based on the high expression level of potential candidate genes, it is speculated that the *QGpc.yzu-2A* region pinpointed one promising candidate gene, TraesCS2A02G473000 (an RNA-binding protein, RBP). The identification of an RBP is particularly intriguing, as it suggests a layer of post-transcriptional regulation in GPC determination. RBPs are known to fine-tune gene expression by influencing mRNA stability, localization, and translation efficiency. This finding opens a new avenue for research, positing that this RBP might regulate the expression of key genes involved in nitrogen remobilization or storage protein synthesis during grain filling. This hypothesis is bolstered by studies in barley, where a putative RBP gene *HvGR-RBP1* was drastically up-regulated in a high-GPC line [18,19]. The co-localization of *QGpc.yzu-6A* with the known *TaNAM-6A* gene not only validates our analysis but also reinforces the critical role of NAC transcription factors in mediating senescence and nutrient remobilization to the grain [9,10,20,21]. The results of this study will be useful for further dissecting the genetic basis of GPC in wheat and developing new wheat cultivars with desirable alleles to improve the stability of grain quality.

While our study provides valuable insights, it is important to consider its limitations. The predominance of the Chinese germplasm in our panel, while valuable for local breeding, may limit the direct transferability of our findings to other gene pools. For instance, prior GWASs in more diverse global panels have identified key loci on chromosomes 2B in European breeding programs, which were not prominent in our study [22]. It is noteworthy that the *QGpc.yzu-6A* locus on chromosome 6A, which was applied in the CIMMYT breeding program, was also detected in our study [23]. This discrepancy underscores the impact of genetic background and highlights the need for validation in a broader germplasm. Furthermore, the candidate genes, including the RBP, require functional validation through approaches like CRISPR-Cas9 gene editing or transgenic complementation to conclusively confirm their roles in regulating GPC.

In conclusion, this study enhances our understanding of the genetic architecture of GPC in wheat through the identification of stable QTNs and biologically relevant candidate genes. The findings provide a valuable resource for marker-assisted selection and genomic breeding strategies. Future work should focus on the functional characterization of the candidate RBP gene and the validation of associated markers in diverse, internationally sourced breeding populations to develop high-quality wheat varieties with stable performance under changing climatic conditions.

## Figures and Tables

**Figure 1 cimb-47-00981-f001:**
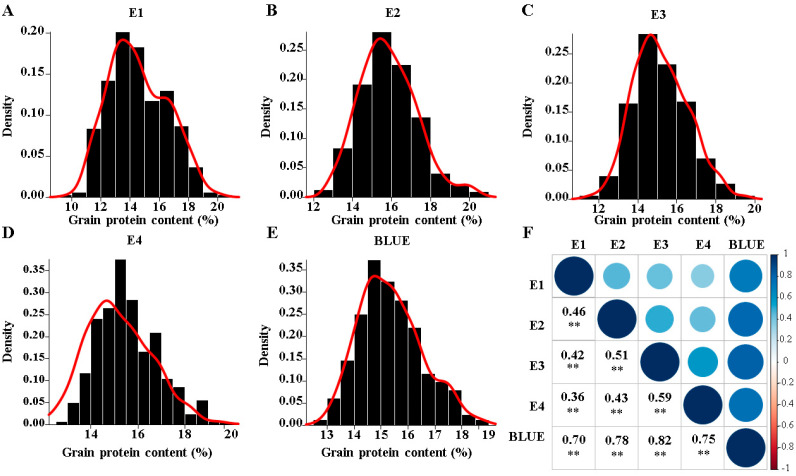
Phenotypic distribution and correlation of the investigated traits in a panel of 327 wheat accessions. (**A**–**E**). Distribution of grain protein content (%) in four environments (E1, E2, E3, E4) and BLUE values. (**F**), Pearson’s product–moment correlation (r) among the investigated traits. ** denotes the significance of *p* < 0.01.

**Figure 2 cimb-47-00981-f002:**
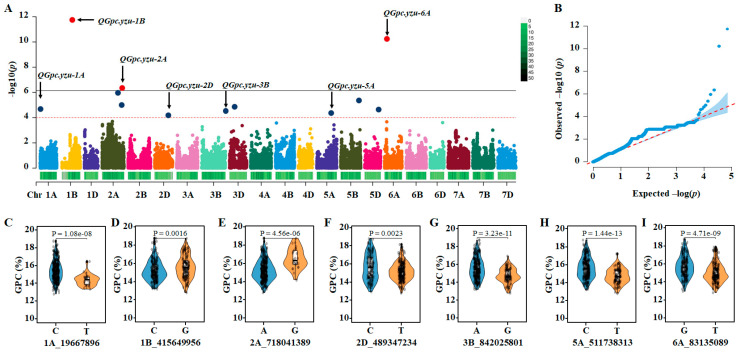
Summary of the genome-wide association studies (GWASs) of GPC in a panel of 327 wheat accessions. (**A**), Manhattan plots show the distribution of marker significance −*log10*(*P*) along wheat chromosomes. Dashed and solid horizontal lines denote the suggestive threshold value (*p* < 1.0 × 10^−4^) and significance threshold value (*p* < 7.20 × 10^−7^), respectively. (**B**), Quantile–quantile plots show the distribution of observed vs. expected (red dashed line) −*log10*(*P*). (**C**–**I**), Analysis of GPC for wheat accessions stratified by genotype at seven significant SNPs. Statistical significance was determined using a two-tailed *t*-test.

**Figure 3 cimb-47-00981-f003:**
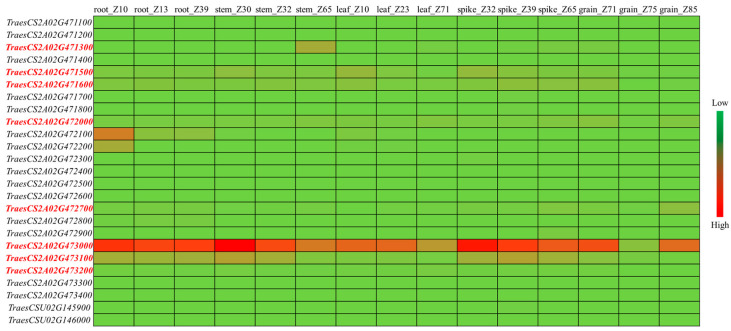
Expression analysis of annotated genes in the *QGpc.yzu-2A* region. The gene ID marked with red color indicates that the expressed genes were detected in the grains. Expression data were sourced from the wheat expression database [17].

**Table 1 cimb-47-00981-t001:** Summary statistics for grain protein content (GPC) in a set of 327 wheat accessions evaluated over two growing seasons.

Trait	Env. ^1^	Mean *	Std. Dev.	Min.	Max.	CV%	*h* ^2^
GPC (%)	E1	14.56 a	2.00	9.87	20.12	13.77	0.26
	E2	15.85 d	1.47	12.81	20.54	9.26	0.49
	E3	15.18 b	1.39	11.46	19.53	9.18	0.20
	E4	15.60 c	1.28	12.99	19.89	8.23	0.33
	BLUE	15.37	1.17	12.82	18.79	7.58	0.52

^1^ E1, E2, E3, and E4 represent the 2023–2024 and 2024–2025 cropping seasons at the Zhenjiang and Yancheng site, respectively. * Different letters denote significant difference between the mean values at *p* < 0.05. Env. = environments; Std. Dev. = standard deviation; CV = coefficient of variation; *h*^2^ = broad-sense heritability; BLUE = best linear unbiased estimator.

**Table 2 cimb-47-00981-t002:** Factorial analysis of variance (ANOVA) for GPC across four environments (growing seasons) for a set of 327 wheat accessions.

Source of Variation	SS	df	MS	F	*p*-Value	F Crit
Genotype (G)	3291.177	326	10.096	8.936	0.000	1.149
Environment (E)	429.863	3	143.288	126.823	0.000	2.606
G × E	2189.864	968	2.262	2.002	0.000	1.113
Total	6238.638	1297				

SS = sum of squares; df = degrees of freedom; MS = mean square; F = F value; F Crit = F critical value.

**Table 3 cimb-47-00981-t003:** QTLs identified for grain protein content (*QGpc.yzu*).

*QTLs*	Chr.	Position (bp) ^1^	QTNs	−*log10* (*P*)	E
*QGpc.yzu-1A*	1A	19,667,896	1A_19667896	4.7	BLUE
				4.3	E3
*QGpc.yzu-1B*	1B	415,649,956	1B_415649956	11.7	BLUE
				4.7	E4
*QGpc.yzu-2A*	2A	718,041,389	2A_718041389	6.3	BLUE
				4.3	E3
				6.1	E4
*QGpc.yzu-2D*	2D	489,347,234	2D_489347234	4.2	BLUE
				4.4	E3
*QGpc.yzu-3B*	3B	842,025,801	3B_842025801	4.4	E3
				4.7	E4
*QGpc.yzu-5A*	5A	511,738,313	5A_511738313	4.4	BLUE
				10.2	E2
*QGpc.yzu-6A*	6A	83,135,089	6A_83135089	10.2	BLUE
				4.2	E4

^1^ Physical position of the markers based on wheat RefSeq v2.1.

**Table 4 cimb-47-00981-t004:** Functional annotation, haplotype structure, and nucleotide diversity within the haplotype block of the *QGpc.yzu-2A* base on the wheat pangenome [16].

Gene ID ^1^	Functional Annotation	H	SS	NS	Pi
TraesCS2A02G471300	Benzyl alcohol O-benzoyltransferase	3	0	6	0.00072
TraesCS2A02G471500	Homoserine kinase	2	1	2	0.00055
TraesCS2A02G471600 ^α^	Phospholipase A2	NA	NA	NA	NA
TraesCS2A02G472000	IMPACT family member in pol 5′region	2	1	2	0.00083
TraesCS2A02G472700	BTB/POZ and MATH domain-containing protein 2	2	3	2	0.00096
TraesCS2A02G473000	RNA-binding family protein	2	0	0	0.00843
TraesCS2A02G473100	Polyadenylate-binding protein-interacting protein 4	3	0	0	0.01172
TraesCS2A02G473200 ^α^	AT hook motif DNA-binding family protein	NA	NA	NA	NA

^1^ Gene ID retrieved from wheat RefSeq v2.1; H is the number of haplotypes; SS is the number of synonymous substitutions; NS is the number of non-synonymous substitutions; Pi is the nucleotide diversity. α represents no variation among the observed varieties. When multiple transcripts were annotated, the number of mutations is indicated if different.

## Data Availability

The original contributions presented in this study are included in the article/Appendix A. Further inquiries can be directed to the corresponding authors.

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
