# Peer review of "GWAS Reveals Stable Genetic Loci and Candidate Genes for Grain Protein Content in Wheat"

_cimb, 2025, doi:10.3390/cimb47120981_

Round 1
Reviewer 1 Report
Comments and Suggestions for Authors
Summary
This authors present a genome-wide association study (GWAS) investigating the genetic basis of grain protein content (GPC) in a diverse panel of 327 wheat accessions. The study identifies seven stable quantitative trait nucleotides (QTNs) across multiple environments, highlights two promising candidate genes within a major QTL on chromosome 2A, and provides evidence for the involvement of TaNAM-6A. The topic is timely and relevant to molecular breeding and crop genomics, and the results have potential utility in marker-assisted selection for wheat quality improvement.
Overall, the manuscript is well-organized and clearly written. The experimental design, statistical analyses, and interpretations are mostly sound. However, the manuscript requires minor revision. Specific issues are outlined below.
Comments
- Experimental design:
The number of environments (four) is acceptable but modest. Consider reporting the environmental parameters (e.g., temperature, rainfall, soil nitrogen) to better explain the observed G×E interaction. - Validation of the candidate genes:
The identification of two candidate genes (TraesCS2A02G473000 and TraesCS2A02G473100) is interesting. However, the functional relevance of these genes to protein metabolism or nitrogen remobilization in wheat is speculative. The discussion should include literature connecting RNA-binding or polyadenylate-binding proteins to grain protein accumulation. - Discussion:
The discussion largely reiterates results. The authors should expand on biological implications. For example, how these QTNs could be integrated into marker-assisted selection (MAS) or genomic prediction frameworks.
Include a short paragraph on limitations and future work (functional validation, larger germplasm panels). The study would benefit from a comparison with prior GWAS for GPC in other global germplasm (e.g., CIMMYT or European panels).
Author Response
Comment 1: Experimental design: The number of environments (four) is acceptable but modest. Consider reporting the environmental parameters (e.g., temperature, rainfall, soil nitrogen) to better explain the observed G×E interaction.
Response 1: Thank you for pointing this out. We have submitted the temperature, rainfall data in Zhenjiang and Yancheng site across two growing seasons (November–May of 2023–2024 and 2024–2025) (Figure S1). The total nitrogen (N) application rate was 225 kg·ha⁻², with 50%, 30%, and 20% applied as base fertilizer, tillering fertilizer, and jointing fertilizer, respectively. The application rates of phosphorus (P₂O₅) and potassium (K₂O) were both 144 kg·ha⁻², with 50% applied as base fertilizer and 50% as jointing fertilizer (line 79-90 in the revised manuscript).
Comment 2: Validation of the candidate genes. The identification of two candidate genes (TraesCS2A02G473000 and TraesCS2A02G473100) is interesting. However, the functional relevance of these genes to protein metabolism or nitrogen remobilization in wheat is speculative. The discussion should include literature connecting RNA-binding or polyadenylate-binding proteins to grain protein accumulation.
Response 2: We agree with this comment. Thank you for pointing this out. We have revised this part as follow: Based on the high expression level of potential candidate genes, it is speculated that the QGpc.yzu-2A region pinpointed one promising candidate genes, TraesCS2A02G473000 (an RNA-binding protein, RBP). The identification of an RBP is particularly intriguing, as it suggests a layer of post-transcriptional regulation in GPC determination. RBPs are known to fine-tune gene expression by influencing mRNA stability, localization, and translation efficiency. This finding opens a new avenue for research, positing that this RBP might regulate the expression of key genes involved in nitrogen remobilization or storage protein synthesis during grain filling. This hypothesis is bolstered by studies in barley, where a putative RBP gene HvGR-RBP1 was drastically up-regulated in a high-GPC line [18,19]. The co-localization of QGpc.yzu-6A with the known TaNAM-6A gene not only validates our analysis but also reinforces the critical role of NAC transcription factors in mediating senescence and nutrient remobilization to the grain [9,10,20,21]. The results of this study will be useful for further dissecting the genetic basis of GPC in wheat and developing new wheat cultivars with desirable alleles to improve the stability of grain quality (line 261-275 in the new revised manuscript).
Comment 3: Discussion: The discussion largely reiterates results. The authors should expand on biological implications. For example, how these QTNs could be integrated into marker-assisted selection (MAS) or genomic prediction frameworks. Include a short paragraph on limitations and future work (functional validation, larger germplasm panels). The study would benefit from a comparison with prior GWAS for GPC in other global germplasm (e.g., CIMMYT or European panels).
Response 3: Agree. We have revised this part in the discussion part in the new revised manuscript. Mention exactly where in the revised manuscript this can be found in line 251-286. The detail information was as follow:
Using the FarmCPU model, we identified seven QTNs associated with GPC. The stability of QGpc.yzu-2A across two environments and the major effect of QGpc.yzu-1B make them prime targets for breeding applications. To translate these findings into practice, the flanking SNPs of these stable QTNs, particularly QGpc.yzu-2A and QGpc.yzu-1B, can be developed into robust Kompetitive Allele-Specific PCR markers. These markers would enable efficient marker-assisted selection to pyramid high-GPC alleles into elite breeding lines, thereby breaking the historical yield-protein trade-off. Furthermore, incorporating these QTNs as fixed effects in genomic prediction models could enhance the accuracy of selecting for GPC, especially when breeding for performance stability across diverse environments.
Based on the high expression level of potential candidate genes, it is speculated that the QGpc.yzu-2A region pinpointed one promising candidate genes, TraesCS2A02G473000 (an RNA-binding protein, RBP). The identification of an RBP is particularly intriguing, as it suggests a layer of post-transcriptional regulation in GPC determination. RBPs are known to fine-tune gene expression by influencing mRNA stability, localization, and translation efficiency. This finding opens a new avenue for research, positing that this RBP might regulate the expression of key genes involved in nitrogen remobilization or storage protein synthesis during grain filling. This hypothesis is bolstered by studies in barley, where a putative RBP gene HvGR-RBP1 was drastically up-regulated in a high-GPC line [18,19]. The co-localization of QGpc.yzu-6A with the known TaNAM-6A gene not only validates our analysis but also reinforces the critical role of NAC transcription factors in mediating senescence and nutrient remobilization to the grain [9,10,20,21]. The results of this study will be useful for further dissecting the genetic basis of GPC in wheat and developing new wheat cultivars with desirable alleles to improve the stability of grain quality.
While our study provides valuable insights, it is important to consider its limitations. The predominance of Chinese germplasm in our panel, while valuable for local breeding, may limit the direct transferability of our findings to other gene pools. For instance, prior GWAS in more diverse, global panels have identified key loci on chromosomes 2B in European breeding programs, which were not prominent in our study [22]. It is noteworthy that the QGpc.yzu-6A loci on chromosome 6A, which was applied in the CIMMYT breeding program, was also detected in our study [23]. This discrepancy underscores the impact of genetic background and highlights the need for validation in broader germplasm. Furthermore, the candidate genes, including the RBP, require functional validation through approaches like CRISPR-Cas9 gene editing or transgenic complementation to conclusively confirm their roles in regulating GPC.
Reviewer 2 Report
Comments and Suggestions for Authors
The manuscript is interesting and could be a good addition to the pool of knowledge. I hereby suggest some major revisions before formal acceptance. Kindly find below:
- Please correct and clarify the broad-sense heritability formula.
- Revise the Materials and Methods to explicitly state the experimental design, specify the exact number of replications per environment, and justify the statistical models used for BLUEs and ANOVA.
- Provide a robust justification for using only the FarmCPU model. Validate the GWAS results by comparing them with at least one other well established standard method viz. MLM, EMMAX).
- Report -log10(P) values for QTNs alongside LOD scores to conform to GWAS standards, with a clear justification if LOD is preferred.
- Move beyond speculation for the candidate genes (TraesCS2A02G473000 and TraesCS2A02G473100). Strengthen their candidacy by analyzing/discussing their expression patterns during grain development (using public transcriptome data if possible) and linking their known biological functions (as RNA-binding proteins) to GPC.
- Deepen the discussion. Specifically, elaborate on the implications of identifying the known gene TaNAM-6A, including whether it validates the approach and the haplotype variation within the panel.
- Clearly discuss the limitation of the panel heavy bias towards Chinese accessions and how this might affect the generalizability of the QTNs for global breeding.
- Briefly discuss the potential environmental factors like nitrogen, temperature, water stress that could explain the substantial differences in mean GPC between the two trial locations (E1 and E2).
I hope these suggestions will enhance the manuscript strength and quality. I am looking forward to see the revised version.
Author Response
Comment 1: Please correct and clarify the broad-sense heritability formula.
Response 1: Agree. We have revised this part as follows:
Broad-sense heritability (h²) was estimated to assess the degree to which GPC is under genetic control. Where VG is the genotypic variance and VE is the environmental variance. This estimation is based on variance components derived from the ANOVA and reflects the proportion of total phenotypic variance that is heritable, indicating the potential for phenotypic selection on this trait. Broad sense heritability h2 for GPC was calculated using the formula:
Mention exactly where in the revised manuscript this can be found in line 117-123.
Comment 2: Revise the Materials and Methods to explicitly state the experimental design, specify the exact number of replications per environment, and justify the statistical models used for BLUEs and ANOVA.
Response 2: Agree. We have revised this part in the new manuscript. Mention exactly where in the revised manuscript this can be found in line 77-90, line 100-116.
Comment 3: Provide a robust justification for using only the FarmCPU model. Validate the GWAS results by comparing them with at least one other well established standard method viz. MLM, EMMAX).
Response 3: We agree with this comment. Thank you for pointing this out. We discussed in detail the rationale for using the FarmCPU model for GWAS, which were found in line 135-143. The detail information was as follows: The FarmCPU model was selected as the sole method for this genome-wide association study due to its demonstrated superiority in effectively controlling false positives while maintaining high statistical power. Unlike the Mixed Linear Model (MLM), which can over-correct for population structure and lead to false negatives, or the fixed effect model (e.g., stepwise regression), which is prone to false positives due to confounding between population structure and kinship, FarmCPU iteratively employs a Fixed Effect Model and a Random Effect Model. This innovative approach effectively separates the testing for marker-trait associations from the control of population structure and kinship.
The MLM model was used for GWAS anslysis, the Manhattan plots were found in Figure S2. We have discussed two model result in the Discussion part.
Comment 4: Report -log10(P) values for QTNs alongside LOD scores to conform to GWAS standards, with a clear justification if LOD is preferred.
Response 4: Agree. Thank you for pointing this out. We have changed LOD to -log10(P) in the new manuscript.
Comment 5: Move beyond speculation for the candidate genes (TraesCS2A02G473000 and TraesCS2A02G473100). Strengthen their candidacy by analyzing/discussing their expression patterns during grain development (using public transcriptome data if possible) and linking their known biological functions (as RNA-binding proteins) to GPC.
Response 5: Agree. We have analyzed the wheat expression database, and found that the expression of gene TraesCS2A02G473000 was markedly abundant in the analyzed tissues, especially in grains, relative to other genes (Figure 3). And the possible explanations were suggested in the discussion part (line 262-271).
Comment 6: Deepen the discussion. Specifically, elaborate on the implications of identifying the known gene TaNAM-6A, including whether it validates the approach and the haplotype variation within the panel.
Response 6: According to the reviewer’s suggestion, we have added some sentence as follow: The co-localization of QGpc.yzu-6A with the known TaNAM-6A gene not only validates our analysis but also reinforces the critical role of NAC transcription factors in mediating senescence and nutrient remobilization to the grain (line 271-273). Haplotype analysis of TaNAM-6A gene has been analyzed by using a collection of 302 bread wheat accessions originating from China [Meng et al., 2024]. Thus, the work of haplotype analysis was not further repeated in this study.
Comment 7: Clearly discuss the limitation of the panel heavy bias towards Chinese accessions and how this might affect the generalizability of the QTNs for global breeding.
Response 6: According to the reviewer’s suggestion, we have revised this as follows in the revised manuscript: While our study provides valuable insights, it is important to consider its limitations. The predominance of Chinese germplasm in our panel, while valuable for local breeding, may limit the direct transferability of our findings to other gene pools. For instance, prior GWAS in more diverse, global panels have identified key loci on chromosomes 2B in European breeding programs, which were not prominent in our study [22]. It is noteworthy that the QGpc.yzu-6A loci on chromosome 6A, which was applied in the CIMMYT breeding program, was also detected in our study [23]. This discrepancy underscores the impact of genetic background and highlights the need for validation in broader germplasm. Furthermore, the candidate genes, including the RBP, require functional validation through approaches like CRISPR-Cas9 gene editing or transgenic complementation to conclusively confirm their roles in regulating GPC. This sentence was found in line 277-287.
Comment 8: Briefly discuss the potential environmental factors like nitrogen, temperature, water stress that could explain the substantial differences in mean GPC between the two trial locations (E1 and E2).
Response 8: Agree. Thank you for pointing this out. We have added the nitrogen, temperature and rainfall data in the Materials and Methods part.